# Bioengineering for the Microbial Degradation of Petroleum Hydrocarbon Contaminants

**DOI:** 10.3390/bioengineering10030347

**Published:** 2023-03-10

**Authors:** Minzhen Wang, Mingzhu Ding, Yingjin Yuan

**Affiliations:** 1Frontiers Science Center for Synthetic Biology and Key Laboratory of Systems Bioengineering (Ministry of Education), Tianjin University, Tianjin 300072, China; 2School of Chemical Engineering and Technology, Tianjin University, Tianjin 300072, China

**Keywords:** biodegradation, n-alkanes, alkane hydroxylases, engineered microbial chassis, microbial community

## Abstract

Petroleum hydrocarbons are relatively recalcitrant compounds, and as contaminants, they are one of the most serious environmental problems. n-Alkanes are important constituents of petroleum hydrocarbons. Advances in synthetic biology and metabolic engineering strategies have made n-alkane biodegradation more designable and maneuverable for solving environmental pollution problems. In the microbial degradation of n-alkanes, more and more degradation pathways, related genes, microbes, and alkane hydroxylases have been discovered, which provide a theoretical basis for the further construction of degrading strains and microbial communities. In this review, the current advances in the microbial degradation of n-alkanes under aerobic condition are summarized in four aspects, including the biodegradation pathways and related genes, alkane hydroxylases, engineered microbial chassis, and microbial community. Especially, the microbial communities of “Alkane-degrader and Alkane-degrader” and “Alkane-degrader and Helper” provide new ideas for the degradation of petroleum hydrocarbons. Surfactant producers and nitrogen providers as a “Helper” are discussed in depth. This review will be helpful to further achieve bioremediation of oil-polluted environments rapidly.

## 1. Introduction

Petroleum hydrocarbons are common environmental pollutants, which endanger terrestrial and aquatic ecosystems due to their sediment and secondary release in the coastal environment [1,2]. Some studies have shown that many plants, even some edible plants, can take up petroleum hydrocarbons from contaminated soil and aqueous media, which is harmful to human health [3]. Traditional physicochemical treatments are expensive and have limited efficiency [4]. With the development of synthetic biology and metabolic engineering strategies, microbial remediation technology was recognized to be one of the most effective approaches to deal with petroleum pollution. Although microbial remediation technology has developed rapidly and made remarkable achievements, there are still some limitations to environmental remediation.

Petroleum hydrocarbons are categorized as n-alkanes, iso-alkanes, cycloalkanes, and aromatics [5], among which n-alkanes are the most biodegradable structural group. However, at physiological temperatures, the C_5_–C_10_ homologs tend to disrupt the lipid membrane structures of microorganisms, and C_20_–C_40_ are hydrophobic solids, which are not easily degraded. As a result, n-alkanes have been detected in lakes, rivers, oceans, groundwater, and soil [6]. Here, gaseous alkanes (<C_5_) are excluded from our consideration due to their physical properties at physiological temperatures, which mean they have very few toxic effects on the environment. For brevity, n-alkanes will be used to represent nongaseous n-alkanes (>C_5_) in the following.

In recent years, microbial remediation technology of petroleum hydrocarbon pollution has been extensively studied from different levels. At the external environment level, the toxicity of petroleum hydrocarbon and factors influencing the microbial degradation of petroleum hydrocarbons have been discussed many times [4,7]. The iron-assisted anaerobic hydrocarbon degradation was discussed separately [8]. Some related technologies have been studied and summarized, such as bioelectrochemical systems [9] and rhizoremediation [10]. The strategies for petroleum hydrocarbon bioremediation in the marine environment were reviewed [11]. At the community level, electron transfer between bacterial cells, microbial interactions, and syntrophic phenomenon during hydrocarbon biodegradation have been well summarized [6]. Here, the idea of microbial community was often mentioned. At the cell level, the mechanism of petroleum hydrocarbon biodegradation under aerobic and anaerobic condition was reviewed eight years ago [12]. A recent review systematically summarized the enzymes and corresponding genes involved in the microbial petroleum degradation pathway [13]. The present review provides a detailed description of alkane hydroxylase, which catalyzes the rate-limiting step, and other genes that contribute to degradation have also been summarized. With the development of synthetic biology and metabolic engineering, it has gradually become possible to construct microbial communities for the degradation of petroleum hydrocarbons by bottom-up approaches. The application of synthetic biology and metabolic engineering in the degradation of petroleum hydrocarbons has been discussed [14]. However, the research basis required for the bottom-up construction of degrading strains and microbial communities is rarely summarized systematically.

Therefore, this review summarized the research basis of constructing artificial systems for petroleum hydrocarbon degradation from four aspects, including the biodegradation pathways and related genes, alkane hydroxylases, engineered microbial chassis, and microbial community. First, this review comprehensively summarizes biodegradation pathways, related genes, and alkane hydroxylase, taking the example of n-alkanes under aerobic conditions. Then, the potential chassis for the construction of degrading strains is discussed. Finally, a new idea of “Alkane-degrader and Alkane-degrader” and “Alkane-degrader and Helper” for constructing microbial communities is proposed. This work will help later researchers to construct degrading strains and then construct microbial communities more efficiently.

## 2. Biodegradation Pathways and Related Genes

### 2.1. Biodegradation Pathways

There are mainly four pathways for n-alkanes biodegradation, including terminal oxidation pathway, subterminal oxidation pathway, diterminal oxidation pathway, and Finnerty pathway. Hexadecane was taken as an example here to explain the four degradation pathways (Figure 1).

Terminal oxidation pathway is the most common degradation pathway and can be found in many bacteria, such as *Alcanivorax borkumensis* SK2(T) [15], *Pseudomonas putida* KT2440 [16], and *Geobacillus thermodenitrificans* NG80-2 [17]. Taking hexadecane as an example, hexadecane is oxidized to 1-hexadecanol by alkane hydroxylases (e.g., AlkB, AlkM, LadA and cytochrome P450 family), and then 1-hexadecanol is further oxidized by alcohol dehydrogenases to 1-hexadecanal [18]. Finally, it is converted by aldehyde dehydrogenases into hexadecanoic acid. As a kind of fatty acid, hexadecanoic acid enters β-oxidation in the end. Subterminal oxidation was found by Forney and Markovitz [19] in *Pseudomonas aeruginosa*. Thirty-three years later, this pathway was recognized in *Gordonia* sp. strain TY-5 [20]. Different from the terminal oxidation pathway, hexadecane is oxidized to 2-hexadecanol by alkane hydroxylases (e.g., AlkB and the cytochrome P450 family). Then, 2-hexadecanol is oxidized to 2-hexadecanal. Under the catalysis of Baeyer–Villiger monooxygenase (BVMO), 2-hexadecanal could be converted into the tetradecyl acetate. The tetradecyl acetate is further oxidized by esterase to the corresponding alcohol, and the next steps are the same as for the terminal pathway. In the end, fatty acids enter β-oxidation. The diterminal oxidation pathway was first discovered in a bacterium. This pathway was also found in several strains of yeast [21]. The special point of this pathway is the existence of ω-hydroxylase, which can convert hexadecanoic acid to more valuable 16-hydroxyhexadecanoic acid. 16-Hydroxyhexadecanoic acid is further oxidized by alcohol dehydrogenases and aldehyde dehydrogenases. The product, hexadecanedioic acid, enters β-oxidation. The Finnerty way is different from the three above-mentioned ones. The pathway was postulated by Finnerty and was found in *Acinetobacter* sp. strain HO1-N [22]. In the first step, hexadecane is catalyzed by dioxygenase and is converted into 1-hydroperoxy hexadecane and hexadecaneperoxoic acid. Then, hexadecaneperoxoic acid is further oxidized to hexadecanoic acid, which enters β-oxidation.

### 2.2. Related Genes

In microorganisms, alkane degradation involves many degradation pathway genes, and it can be impacted by many other processes, such as oxidative stress reaction and recognition and transport processes. Therefore, it is necessary to consider all the genes involved when constructing and optimizing an alkane degrader. With the development of synthetic biology and metabolic engineering strategies, many genes related to the degradation of alkanes have been identified and analyzed. Some important genes are as follows (Figure 1).

In the process of alkane degradation, the genes encoding alkane hydroxylase play an important role. Some genes, such as *alkB* [23], *alkM* [24], *ladA* [25], and the gene of encoding CYP153C1 protein [26] have been identified in prokaryotes. Meanwhile, *rubA* and *rubB* [27] were recognized to encode alkane hydroxylase coenzymes in prokaryotes, and *alkR*, which is located next to *alkM* with an opposite orientation, encodes a transcriptional regulator of polypeptide. In the alkane degradation of eukaryotes, *CYP52* gene family occur frequently, especially in yeast [28,29]. For example, twelve genes (*ALK1* to *ALK12*) which encode the cytochromes P450ALKs have been isolated in *Yarrowia lipolytica* [30]. Furthermore, *CYP52A21* and *CYP52A23* genes, which belong to *CYP52* family, encode an alkane/fatty acid hydroxylase. It can not only oxidize alkane but also ω-hydroxylate fatty acid in *Candida albicans* [31]. In the degradation of alcohols, *laoA* and *laoB* encode alcohol dehydrogenase and its coenzyme, respectively, to degrade primary long-chain alcohols; then, *laoC* encodes an aldehyde dehydrogenase, converting long-chain aldehyde to the corresponding acids [32]. In addition, *HFD1-HFD4,* which was identified from *Y. lipolytica* also can perform the same function as *laoC* in the alkane degradation pathway [33]. On a secondary alcohol degradation gene cluster, *sadC* is involved in the degradation of secondary alcohol; *sadD, sadA,* and *sadB* encode a BVMO, and two esterases catalyze secondary alcohol to the corresponding primary alcohols [34]. In the end, acyl-CoA synthetase encoded by *FAT1* and *FAA1* catalyzes fatty acid to enter the TCA cycle [35].

The recognition and transport of alkanes and the oxidative stress reaction are two important aspects that impact the degradation of alkanes. The recognition and transport of alkanes usually takes place on the cell membrane. Genes *aupA* and *aupB* which encode outer membrane proteins (AupA) and inner membrane proteins (AupB) respectively are responsible for the uptake of alkanes in a prokaryote [36]. Gene *ABC1*, which encodes the ABC1 transporter is involved in the transport of alkanes in a eukaryote [37]. Some genes that encode fatty acid transport proteins, such as *ScFAT1* and *YlFAT1* [38] in eukaryotes and *exfadLO* [39] and *fadL* [40] in prokaryotes, may indirectly impact alkane degradation by increasing or decreasing fatty acid accumulation. In addition, three genes, *Yas1p*, *Yas2p*, and *Yas3p*, which encode an alkane-responsive biosensor, were transferred artificially to *Saccharomyces cerevisiae* [41]. This study has enlightening significance for the construction of *S. cerevisiae* that can degrade alkanes. In the process of alkane degradation, some harmful substances such as H_2_O_2_, O_2_^−^, and organic sulfonates may be produced to cause oxidative damage to cells. At the same time, related genes have been identified. *oxyR* which encode the virulence-related redox-sensing transcription factor confers resistance to H_2_O_2_ [42]. The *P24* gene was recognized to encode a superoxide dismutase, which can protect from damage due to O_2_^−^ [43]. *ssuD* and *tauD* can transform organic sulfonate to a sulfur source, which is necessary for the production of some oxidative stress-sensing proteins and metabolites that defend against oxidative stress [44]. It is worth mentioning that Osh6p which be encoded by *OSH6* gene is a homologue of the oxysterol-binding protein. It was speculated to be involved in the formation of Alk protein by changing the ratio of phosphatidylserine and then impacting the endoplasmic reticulum membrane environment [45].

## 3. Alkane Hydroxylases

In alkane degradation, alkane hydroxylases participate in the first reaction by introducing oxygen atoms, which be considered as the rate-limiting step [22]. At present, the alkane hydroxylases including integral-membrane alkane hydroxylases (e.g., AlkB, AlkM), cytochrome P450 alkane hydroxylases, flavoprotein alkane hydroxylases (e.g., AlmA, LadA), and dioxygenase have been identified. Among them, the research on integral-membrane alkane hydroxylases and cytochrome P450 alkane hydroxylases is more mature and detailed than that on flavoprotein alkane hydroxylases and the dioxygenase family in the aspects of structure and modification (Table 1).

### 3.1. Integral-Membrane Alkane Hydroxylases

AlkB and AlkM, which were identified from *P. putida* Gpo1 [46] and *Acinetobacter calcoaceticus* ADP1 [24], respectively, were the two major enzyme systems. The difference of function between them is that AlkM was reported to oxidate long-chain n-alkanes (C_16_–C_30_) [47] and AlkB oxidates alkanes with shorter chain lengths (C_5_–C_12_) [48].

Compared to AlkM, the structure of AlkB has been studied more clearly. In a previous study, AlkB was studied to have six alpha-helical transmembrane segments, and each transmembrane segment had eight histidines. These eight histidines are very significant for the structure of AlkB [49]. In addition to this, the existence of iron is crucial for the activity of AlkB. In *P. putida* Gpo1, the alkane hydroxylase system is composed of many enzymes involved in alkane degradation. In addition to AlkB, there are also two rubredoxins and one rubredoxin reductase, which play the role of transferring electrons in the system [48]. Researchers also found another rubredoxin and rubredoxin reductase, rubA and rubB, which are from *A. borkumensis* [50], *Acinetobacter* sp. M-1 [27]. In *Dietzia* strain DQ12-45-1b, researchers found a novel gene *alkW*_1_, which is composed by AlkB and a rubredoxin [51]. The study found the AlkB-fused rubredoxin can oxidate long-chain alkanes. In addition, the *alkB* gene promoter (P*_alkB_*), which impacts the expression of *alkB*, was identified and analyzed [52].

Gene *alkM* is different from *alkB* in DNA sequence homology. However, AlkM is still recognized to belong to integral-membrane alkane hydroxylases for it also needs rubredoxin and rubredoxin reductase [53]. The gene of AlkM also differs from AlkB in its positional relationship with the gene of coenzyme. *AlkM* is not linked to the rubredoxin- and rubredoxin reductase-encoding genes on the *Acinetobacter* sp. strain ADP1, while *alkB* is the opposite of *alkM* [24]. In addition, the transcriptional regulators have also been identified. AlkR is essential for the transcriptional regulation of AlkM. The gene of AlkR is located next to the gene of AlkM [24]. In *Acinetobacter* sp. strain M-1, there are genes *alkMa* and *alkMb*, *alkRa* and *alkRb*, which are similar to strain ADP1 in the aspects of sequence homology and positional relationship [27].

### 3.2. Cytochrome P450 Alkane Hydroxylases

Many alkane hydroxylases belong to the cytochrome P450 family, which are heme-thiolate proteins [54]. Currently, the most studied is Class I and Class II of cytochrome P450, which is categorized by components [55]. CYP153 enzymes was identified from *A. calcoaceticus* EB104 [54], which are an important part of Class I. n-Alkanes (C_6_–C_11_) can be oxidized by CYP153 enzymes, such as CYP153C1, which was cloned from *Novosphingobium aromaticivorans* DSM12444 [26]. Understanding these enzymes can help us modify them in the desired direction with synthetic biology and metabolic engineering strategies. The P450cam system (belonging to CYP101) identified from *P. putida* ATCCI7453 is similar to integral-membrane alkane hydroxylases [56]. The components of the P450cam system include cytochrome P450 alkane hydroxylases, putidaredoxin, and putidaredoxin reductase. The two coenzymes undertake the function of transferring electrons from NADH to P450cam [57]. P450cam has been modified in its active-site amino acid residues by substitution. The mutations have higher activity in alkane degradation [58]. In Class II, the CYP52 family and P450 BM-3 have been studied. CYP52 family is important for ω-oxidation in the diterminal oxidation pathway, which always occurs in yeast such as *Candida tropicalis* ATCC 20336 [59]. In *C. tropicalis* ATCC 20336, the CYP52 family oxidizes n-alkanes to fatty acids and dicarboxylic acids, which are more valuable products. P450 BM-3 is from *Bacillus megaterium* 14581 [60]. The peculiarity of the structure of this enzyme is that the hydroxylase domain and the reductase domain are fused in a single polypeptide chain, which cause it to be the most active P450 enzyme [60]. In addition, some modifications of P450 BM-3 have been studied, such as rational evolution, which improves the degradation ability of n-alkanes [61].

### 3.3. Flavoprotein Alkane Hydroxylases

Flavoprotein alkane hydroxylases catalyze a reaction involving NAD(P)H, cofactor flavin, and substrate. They transfer electrons through redox reactions [22]. At present, two flavoprotein hydroxylases that have been wildly studied are LadA and AlmA. LadA was isolated from *G. thermodenitrificans* NG80-2 [27]. Research showed that it can oxidize long-chain n-alkanes, from C_15_ to at least C_36_ via the terminal oxidation pathway. LadA belongs to the SsuD subfamily, which contains a triosephosphate isomerase (TIM) barrel structure [17]. It was identified to be a flavoprotein hydroxylase for the FMN in the TIM barrel. Some modifications such as random- and site-directed mutagenesis of LadA have been used to enhance its activity [62]. Research showed that the expression of the LadA mutants can help *Pseudomonas fluorescens* strain KOB21 grow faster with hexadecane. AlmA, which was first isolated from *Acinetobacter* sp. strain DSM 17874, can oxidize long-chain alkanes [63]. It was also found in *Alcanivorax dieselolei* B-5 [64], *Acinetobacter pittii* SW-1 [65], *Acinetobacter oleivorans* DR1 [66], *P. aeruginosa* SJTD-1 [67], and *Alcanivorax hongdengensis* A-11-3 [68]. However, few of degradation mechanism that AlmA is involved in have been characterized. In *Acinetobacter* sp. strain M-1, one dioxygenase was found [69]. Taking hexadecane as an example, dioxygenase can oxidate hexadecane to 1-hydroperoxy hexadecane in the Finnerty way. The activity of this enzyme requires flavin adenine dinucleotide and Cu^2+^. The enzyme catalyzes n-alkanes ranging from C_10_ to C_30_. Although there is no genuine repetition of the work that can fully prove dioxygenase acts on n-alkanes in the subsequent 25 years, dioxygenases appeared in the aerobic degradation pathway of naphthalene and phenanthrene during the past 25 years [70,71]. It makes researchers think that the possibility exists, but still needs to be verified. Therefore, the pathway catalyzed by this enzyme is shown by dotted lines in Figure 1.

**Table 1 bioengineering-10-00347-t001:** Some important alkane hydroxylases.

Enzyme	Origin		Structural Features	Modification	Type of Oxidation	Oxidation Length	Reference
The ALKB family	AlkB	*P. putida* GPo1	Six alpha-helical transmembrane segments Nonheme iron integral-membrane Eight histidines Needs iron and oxygen	/	Terminal oxidation, subterminal oxidation	C_5_–C_12_	[48,49]
	AlkM	*Acinetobacter* sp. strain ADP1	/	/	Terminal oxidation	C_16_–C_30_	[24,47]
Cytochrome P450	Class I (CYP153)	Bacteria *A. calcoaceticus* EB104	FAD-containing reductase Iron–sulfur protein	Active site replaced by residues with bulkier and more hydrophobic side chains	Terminal oxidation, subterminal oxidation	C_6_–C_11_	[54,56,72]
	Class II (CYP52)	Fungi *C. tropicalis* ATCC 20336 Bacteria *B. megaterium* 14581	FAD- and FMN-containing cytochrome P450 reductase	Rational evolution	Terminal oxidation, subterminal oxidation	C_10_–C_16_	[59,60,61]
Flavoprotein	LadA	*G. thermodenitrificans* NG80-2	TIM barrel fold C-terminus of polypeptide chain	Random- and site-directed mutagenesis	Terminal oxidation	C_15_–C_36_	[17,25,62]
	AlmA	*Acinetobacter* sp. strain DSM. 17874	Flavin binding	/	/	>C_32_	[63]
	Dioxygenase	*Acinetobacter* sp. strain M-1	ND	/	Finnerty way	C_10_–C_30_	[67]

Note: ND = not defined.

## 4. Engineered Microbial Chassis

Most of identified microorganisms that can express alkane hydroxylases are nonmodel microorganisms such as *Mycobacterium* sp. [73], *Rhodococcus* sp. [74], *Pseudomonas* sp. [75], *Dietzia* sp. [76], *Acinetobacter* sp. [24], *Aspergillus* sp. [73], *Fusarium* sp. [77], *Penicillium* sp. [78], *Ochrobactrum* sp. [79], or *Brevibacterium* sp. [80]. However, they are difficult to modify due to their unstated genetic background. For a microbial chassis, model microorganisms, such as *Escherichia coli* are the most widely used. With the progress of synthetic biology technology, some nonmodel microorganism can also be transformed, and these nonmodel organisms may have functions that model organisms do not have, such as a complete degradation system. Here, we summarize the research progress on *E. coli*, *Pseudomonas* sp., and *Bacillus* sp., which can be modified as potential chassis for the degradation of petroleum hydrocarbons.

### 4.1. E. coli

*E. coli* is an important model microorganism. At present, *E. coli* has been applied to the expression of alkane hydroxylase and the whole-cell biooxidation of alkanes, which is able to not only avoid the complicated steps of enzyme purification but also solve the difficulty of the reduced activity of the enzymes.

In *E. coli*, the research can be divided into two aspects. On the one hand, researchers make *E. coli* degrade alkane by expressing alkane hydroxylases. Back in 1993, researchers tried to introduce *alk* genes into *E. coli*; they found that it resulted in the overexpression of alkane hydroxylase in a distinct cytoplasmic membrane subfraction [81]. Two years later, a study showed that recombinant *E. coli* can express more alkane hydroxylase (AlkB), but the specific activity of the alkane hydroxylase component AlkB was five or six times lower than in *Pseudomonas oleovorans* [82]. However, both studies did not use recombinant *E. coli* to degrade alkane. Recently, a study demonstrated that using the *E. coli* GEC137 pCEc47ΔJ strain to produce primary alcohols and carboxylic acids is feasible [83]. It is the first time using AlkB to degrade n-alkane (n-dodecane) in a recombinant organism. The next year, the AlkB gene of *P. putida* GPo1 was constructed in a PCom8 expression vector, and the pCom8-GPo1 AlkB plasmid was transformed into *E. coli* DH5a [84]. The result showed that the culture of the recombinant *E. coli* with the petroleum biodegradation bacterial community increased the degradation ratio of diesel oil at 24 h from 31% to 50% [84]. On the other hand, other studies focused on ω-oxidation, which can convert alkanes to more valuable chemicals. These studies have shown that diterminal oxidation can be used convert inexpensive medium-chain n-alkanes to valuable medium-chain α,ω-diols and α,ω-dicarboxylic acid (DCAs). In other research, the biocatalytic conversion of fatty acid esters toward ω-hydroxy fatty acid esters was investigated with recombinant *E. coli* that produce the AlkBGT enzymes [85]. Furthermore, a review systematically demonstrated the strategies for improving product yield and productivity of ω-HFAs and their related chemicals in *E. coli* recently [86]. These results indicate that the use of recombinant *E. coli* to degrade alkanes and produce valuable byproducts is feasible.

### 4.2. Pseudomonas sp.

Generally, *Pseudomonas* species are ubiquitous in nature and capable of producing biosurfactants with crude oil as the carbon source. Recently, many strains belonging to *Pseudomonas* sp. were isolated and identified (Table 2). Based on these studies, many genes associated with degradation were identified. For example, a gene cluster encoding a putative alcohol dehydrogenase (PA0364/LaoA), a probable inner membrane protein (PA0365/LaoB), and a presumable aldehyde dehydrogenase (PA0366/LaoC) were explained specifically [32]. Subsequently, modifications of the *Pseudomonas* sp. were also gradually developed. NAH7 plasmid was transferred to *P. putida* KT2440, which enabled *P. putida* KT2440 to degrade naphthalene. At the same time, this transformation alleviated the cellular stress brought on by this toxic compound [87]. This is a rational modification example of *Pseudomonas*. Other research is about irrational modifications of *Pseudomonas*. *Pseudomonas pseudo alcaligenes* CECT 5344 was grown in furfuryl alcohol, furfural, and furoic acid as carbon sources and the evolved strain was obtained after the strain adapted [88]. The evolved strain did not show any prolonged lag phases, while the original strain had a lag period of several days [88]. These studies provide a chassis for the future genetic engineering of wild *Pseudomonas* and laid the foundation for the construction of communities for efficient alkane degradation.

### 4.3. Bacillus sp.

*Bacillus* species have strong resistance to external harmful factors. Therefore, it is appropriate to use *Bacillus* sp. as a chassis for degrading alkanes. They exist in soil, water, air, and animal intestines [109]. Recently, many bacteria of the *Bacillus* sp. were isolated and identified (Table 2). At present, the biological methods of modifying strains mainly include promoter replacement and the strengthening of target genes. In the approach of promoter replacement, through the substitution of the promoter of the lichenysin biosynthesis operon, the engineered strain produced 2149 mg/L lichenysin, a 16.8-fold improvement compared to that of the wild strain [110]. In the approach of the strengthening of target genes, engineered bacteria increase the yield of target products by the activation of two competence-stimulating pheromones to stimulate the transcription of the operon [111], by the independent or simultaneous overexpression of two regulatory genes [112] and by inserting a strong constitutive promoter upstream of the operon [113]. Additionally, Wu et al. [114] developed a systematic engineering approach to improve the biosynthesis. The final surfactant titer increased to 12.8 g/L, with a yield of 65.0 mmol/mol sucrose (42% of the theoretical yield) in the metabolically engineered strain. Although there has been no modification research on the degradation ability of *Bacillus* sp., this still provides some experience for our genetically engineered strains of wild *Bacillus.*

## 5. Microbial Community

In terms of petroleum hydrocarbon degradation, a microbial community is more effective than a single microorganism for it can increase the variety of degradable substrates and construct a system of commensalism and co-metabolism [115,116]. Many microorganisms have been isolated and identified from the environment, and the degradation effects of different combinations, e.g., pure bacterial community [117], pure fungal community [118], bacterial–fungal community [119], and pure protozoa community, have been compared [120]. These undoubtedly provide a rich material and theoretical basis for our further research. However, it is difficult to manipulate microorganisms to do degradation tasks precisely; only if the endogenous gene of each microorganism in microbial communities and cell–cell interaction in communities is known clearly, this can be achieved. Therefore, researchers have begun to consider constructing microbial communities for the degradation of petroleum hydrocarbons designed by using engineered strains and rational allocation of strains. In these studies, the interaction of the microorganisms could be defined as “Alkane-degrader and Alkane-degrader” and “Alkane-degrader and Helper” (Figure 2).

### 5.1. Alkane-Degrader and Alkane-Degrader

In an “Alkane-degrader and Alkane-degrader” community, each member can degrade alkanes, but the type of alkane they degrade is different. Each member degrades an alkane that they are good at degrading, and then they finish the task of degradation of complex petroleum hydrocarbons. Recently, a co-culture of bacteria, *Acinetobacter baumannii*, and fungi, *Talaromyces* sp., was studied. In this community, fungi have stronger ability to degrade n-alkanes, while bacteria degrade other components such as aromatics and iso-alkanes better. The total degradation rate of crude oil of this community can reach 80% [121]. In addition to using wild bacteria to create an artificial community, some researchers tend to use engineered bacteria to construct a microbial community. In one research, a mutant with *alkMa* or *alkMb* deletion of *Acinetobacter venetianus* strain RAG-1 enhanced the degradation of C_10_–C_20_ or C_22_–C_32_ n-alkanes. Then, an adjustable and targeted community consisting of Δ*alkMa/almA* and Δ*alkMb* was constructed. This community achieved enhanced degradation (10 days) of light crude oil (73.42% to 88.65%), viscous crude oil (68.40% to 90.05%), and high waxy crude oil (47.46% to 60.52%) compared with the single wild-type strain [122]. It can be seen that the microbial community in the form of “Alkane-degrader and Alkane-degrader” is conducive to the degradation of complex alkane components.

### 5.2. Alkane-Degrader and Helper

In an “Alkane-degrader and Helper” community, one or some members degrade alkanes, and other member do other works to help the former degrade alkanes better. For example, surfactant producers can help an “Alkane-degrader” degrade alkane by increase the solubility of the alkanes. In contaminated soil, nitrogen providers can help by providing nitrogen to the alkane degrader.

#### 5.2.1. Surfactants Producers

Most “Helpers” can secrete surfactants to help “Alkane-degraders” [123]. However, how can this help the microbes? This question is still is not fully answered. The common explanation for this question is that biosurfactants can increase the solubility, which can result in higher growth of bacteria and greater efficiency of biodegradation [124]. However, more-comprehensive analysis of this process has been carried out taking into account multilevel changes occurring in cells from the genome, through metabolic activity, to the surface properties of the cells [125]. These focused on the in-depth description of relationships between hydrocarbons and bacteria and found changes in the genome caused by exposure to surfactants. The results indicated that the benefits of surfactant use may be related to deep modifications not only of the cell’s surface properties but also at the omics level. Nevertheless, the mechanism of deep modification at the omics levels still needs more research. It should be pointed out that, in some cases, surfactants are not beneficial or may even be harmful to the degradation of petroleum hydrocarbons. In soil pollution remediation, anionic surfactants react with divalent ions such as Ca^2^⁺ or are irreversibly sorbed on the soil. It will result in a large loss of surfactant or increase the soil organic carbon content, which adversely impacts petroleum biodegradation [126]. In addition, some surfactants have been reported to inhibit or stimulate the growth of microorganisms [127].

Based on the differences in their chemical compositions, biosurfactants can be classified into bioemulsifiers, which have higher molecular weights, e.g., polymeric and particulate biosurfactants [128], and biosurfactants, whose molecular weights are lower, e.g., rhamnolipids [129], sophorolipids [130], and lipopeptides [131]. The difference of function between the two types is that low-molecular-weight biosurfactants can help bacteria increase the surface area of hydrophobic, water-insoluble substrates and increase the bioavailability of hydrophobic compounds by reducing the surface tension effectively. However, high-molecular-weight bioemulsifiers coat the oil droplets and prevent their coalescence to achieve this effect [132].

Biosurfactants can increase the degradation rate of alkane degraders [133,134]. Isolation and identification of a surfactant-producing bacteria from the environment has, therefore, become an area of interest for researchers. Understandably, since surfactant production and alkane degradation are closely related, most bacteria that can produce surfactants can also degrade crude oil (Table 3). These studies provide rich materials and theories for the subsequent construction of microbial communities. Additionally, many surfactant producers have been used to increase the degradation rate in a community. The “Helper” could be single bacteria, such as *Pseudomonas* sp. XM-01 [135], *Rhodococcus erythropolis* OSDS1 [124], *B. subtilis* SPB1 [136], *Acinetobacter* sp. Y2 [137], or *Dietzia* sp. CN-3 [138], or a bacterial community, such as cultivable biosurfactant-producing single cultures composed of *Pseudomonas* sp. S2WE, *Pseudomonas* sp. S2WG, *Pseudomonas* sp. S2MS, or *Ochrobactrum* sp. S1MM [139], or a bacterial community. After the “Helper” joins the community, the alkane hydrocarbon degradation rate of all alkane degraders increases significantly, and the highest improvement can reach about 55.4%.

#### 5.2.2. Nitrogen Providers

The importance of nitrogen in the remediation of diesel-contaminated soil is well known [151,152]. Research shows that proper nitrogen biostimulation has a positive effect on the degradation of aromatic hydrocarbons, while excessive nitrogen stimulation has a negative effect on the microbial degradation efficiency [153]. In a community, alkane degraders and nitrogen-fixing bacteria synergistically contribute to each other by providing carbon to the nitrogen-fixing bacteria and nitrogen to the alkane degraders [154]. Researchers found that nitrogen starvation can significantly reduce the strength of cell adhesion to the soil particles [155]. To supplement the nitrogen source required by the alkane degrader, Koren et al. [156] used uric acid bound to crude oil and found that it was available for bacterial growth and petroleum biodegradation. To further explore the role of nitrogen in the alkane degradation, Gao et al. [157] identified the nitrogen metabolic pathway in the bioremediation of diesel-contaminated soil by metagenomics analysis. As a result, they found that, for the best performance of enhanced bioremediation of diesel-contaminated soil, the organic ammonium form of nitrogen is preferred to other form of nitrogen sources. This study will undoubtedly increase our understanding of the mechanism by which nitrogen assists in the degradation of crude oil.

As the importance of nitrogen in alkane degradation is discovered, some strains that can provide nitrogen were also isolated and identified. Recently, a nitrogen-fixing bacterium from an oil production mixture of the Yumen Oilfield was isolated and identified. It belongs to *Azospirillum* [158]. Even before this, scientists have found bacteria with similar functions [159], which can be used to construct communities. Additionally, a genome for novel nitrogen-fixing bacteria with capabilities for the utilization of aromatic hydrocarbons was assembled [160]. This study will open the broad window of bioremediation strategies under a nitrogen-stress environment. In recent years, nitrogen providers were used in communities to increase the degradation rate. For example, one nitrogen-fixing microbe *Azotobacter vinelandii* KCTC2426 helped two different oil-degrading microbes (*Acinetobacter* sp. K-6 + *Rhodococcus* sp. Y2-2) to degrade the diesel from the soil and removed 83.1% of it after 40 days of treatment [161]. In another study, nitrogen-fixing bacteria facilitated the microbial degradation of poly (butylene succinate-co-adipate) by enhancing fungal abundance, accelerating plastic-degrading enzyme activities, and shaping/interacting with plastic-degrading fungal communities [162]. These research results provide great inspiration for us to construct a microbial community with a higher degradation rate in the future.

## 6. Conclusions

Microbial degradation of petroleum has always been a concern of the scientific community. However, the rapid advances of synthetic biology and metabolic engineering strategies in recent years have opened new research avenues on this topic. It is expected that precise and effective degradation of petroleum hydrocarbons can be achieved through the design and construction of degrading strains and microbial communities. The studies on degradation pathway, related genes, enzymes, degrading strain chassis, and the interaction of strains are the basis of realizing the controllable degradation of petroleum hydrocarbon pollutants. The degradation pathways of various components in petroleum hydrocarbons have been fully summarized. However, due to the complexity of microorganisms and the difficulty of obtaining some extremophiles, researchers need to further explore whether there are unknown degradation pathways. There are many studies on genes for the degradation of petroleum hydrocarbons, and more attention should be paid to other genes that impact degradation, which have been summarized in the present review. In the engineered microbial chassis, *S. cerevisiae*, as a model organism of fungi, is also worth considering because of its rich resource environment and advanced molecular manipulation methods. Unfortunately, few studies on the use of this strain for the construction of an alkane degrader were reported. Microbial communities can be used in many scenarios. In the future, the research work of constructing microbial communities to achieve the degradation of petroleum pollution should be conducted more widely.

## Figures and Tables

**Figure 1 bioengineering-10-00347-f001:**
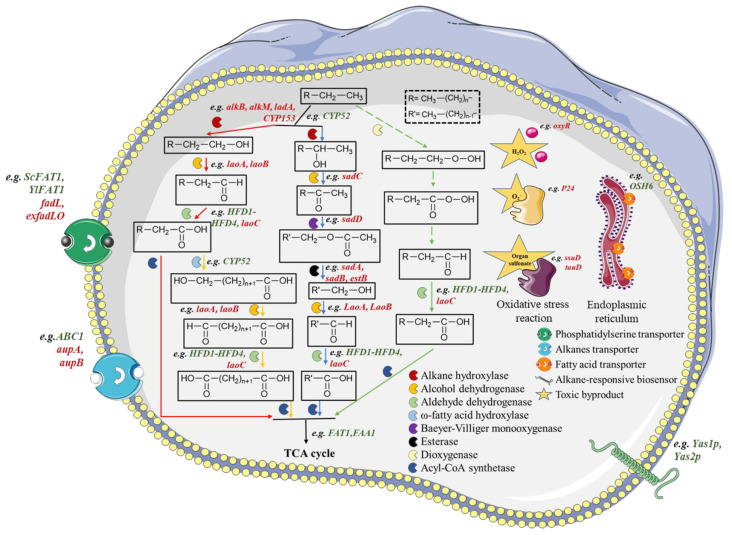
Four pathways for alkane biodegradation. Terminal oxidation, subterminal oxidation, diterminal oxidation, and Finnerty way are indicated by red, blue, orange, and green, respectively. Red italics indicate prokaryotic genes and green italics indicate eukaryotic genes. Dashed lines indicate a putative pathway.

**Figure 2 bioengineering-10-00347-f002:**
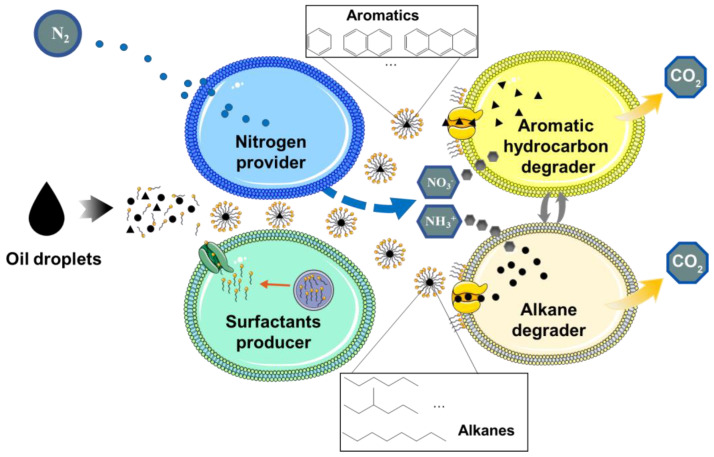
Microbial community for petroleum hydrocarbon contaminant biodegradation.

**Table 2 bioengineering-10-00347-t002:** Some alkane-degrading bacteria that have been recently modified or isolated and identified.

Species	Type	Name	Degrading Substances	Degradation Rate	References
*E. coli*	Artificial	*E. coli* GEC137 pCEc47ΔJ	n-Dodecane	/	[83]
		*E. coli* DH5a pCom8-Gpo1 AlkB	Diesel oil	24 h from 31% to 50%	[84]
*Pseudomonas* sp.	Wild	*Pseudomonas qingdaonensis* ZCR6	Petroleum hydrocarbons	76.52%	[75]
		*P. aeruginosa* pp4	Crude oil	86%	[89]
		*P. aeruginosa* AKS1	Crude oil	0.038 for 1 day	[90]
		*Pseudomonas* sp. strain SA3	Naphthalene	98.74 for 4 days	[91]
		*Pseudomonas brassicacearum* MPDS	Dibenzofuran	65.7% for 4 days	[92]
		*Pseudomonas* sp. *strain* NEE2	n-Hexane	60% for 2 days	[93]
		*Pseudomonas* sp. Sp48	Crude oil	89% for 6 days	[94]
		*P. aeruginosa* L10	C_10_–C_26_ n-alkanes	ND	[95]
		*P. putida* strain KD6	Petroleum hydrocarbons	97.729% for 12 days	[96]
		*P. aeruginosa* strain ASP-53	Pyrene	30.1% after 144 h	[97]
		*P. aeruginosa* ZS1	Crude oil	50% for 12 days	[98]
	Artificial	*P. putida* KT2440R (NAH7)	Naphthalene	/	[87]
		*P. pseudo alcaligenes* CECT 5344 evolved	Furfural and furoic acid	/	[88]
*Bacillus* sp.	Wild	*Bacillus marsiflavi* Bac 144	Crude oil	65% for 5 days	[99]
		*Bacillus* sp. AKS2	Crude oil	0.020 for 1 day	[90]
		*Bacillus subtilis* strain Al-Dhabi-130	Crude oil	89% for 2 days	[100]
		*B. subtilis* RSL-2	Crude oil	ND	[101]
		*Bacillus cereus* T-04	Crude oil	60%–80%	[102]
		*Bacillus safensis* strain ZY16	n-Hexadecane	98.20%	[103]
		*B. subtilis* MG495086	Light paraffin oil	91.3 ± 5%	[104]
		*B. cereus* S13	Anthracene	82.29% for 120 h	[105]
		*Bacillus thuringiensis* AT.ISM.1	Anthracene	91%	[106]
		*Bacillus* spp. B6	PAHs	11%–83%	[107]
		*Bacillus subtilis* (M16K and M19F)	Crude oil	>94.0%	[108]

Note: ND = not defined.

**Table 3 bioengineering-10-00347-t003:** Surfactant-producing bacteria isolated and identified from the environment in recent years.

Name	Culture Conditions	Surfactant Production Capacity	Emulsifying Ability	Degradability	Features	References
*P. aeruginosa* sp. PP4	37 °C, pH 7, MSM broth, 150 rpm	ND	ND	Biodegradation efficiency of crude oil reached 78% for 15 days	Acid tolerant	[140]
*Pseudomonas* sp. *strain* W10	37 °C and 180 rpm	Produced biosurfactant BSW10 (2 g/L)	Reduced the surface tension to 32 mN/m	Degradation of phenanthrene reached 80%	/	[141]
*Rhodotorula* sp. CC01	30 °C, 180 rpm, pH 6.5–7.0 fermentation medium	Production rate: 163.33 mg/L for one hour yield: 1360 mg/L at 50 h	Reduced the surface tension of water to 34.77 ± 0.63 mN/m	Olive oil was determined as the best sole carbon source	Removes NH_4_⁺–N	[142]
*Planococcus* sp. XW-1	pH 7.4, 2216E liquid medium	Glycolipid-type biosurfactant	Reduced the surface tension of water to 26.8 mN/m	After 21 days, 54% of crude oil was degraded	Cold adapted	[143]
*Achromobacter* sp. *A-8*	30 °C, pH 7, and 10 g/L NaCl	ND	Decreased the viscosity of petroleum by about 45%	The biodegradation of petroleum reached 56.23–73.87% for 20 days	Salt tolerant	[144]
*Bacillus licheniformis strain* DM-1	45 °C, LB liquid medium	Exopolysaccharide	Viscosity of the crude oil was reduced by 40%	The degradation of n-octadecane was 81.33%	Tolerates high temperature	[145]
*Geobacillus stearothermophilus* DG1	45–75 °C, fermentation medium	Exopolysaccharide	ND	/	Tolerates high temperature	[146]
*A. pittii* strain ABC	25 ± 2 °C, darkness, 130 rpm	Produced lipopeptide biosurfactant (0.57 g/L)	Emulsification index (E24 65.26 ± 1.2%),	Degraded 88% and 99.8% of n-hexane	Tolerates heavy metal salts	[147]
*Clostridium* sp. N-4	pH 7, 96 °C, 4% salinity	Glycoprotein	Reduced the surface tension of water to 32 ± 0.4 mN/m	ND	Tolerates high temperature	[148]
*Bacillus methylotrophicus* UCP1616	28 °C, pH 7, solid fermentation medium	Concentration of lipopeptide (10.0 g/L)	Reduced the surface tension of water to 29 mN/m	ND	/	[149]
*R. erythropolis* M-25	ND	ND	ND	70.7% of the crude oil was degraded after 30 days	/	[150]

Note: ND = not defined.

## Data Availability

Not applicable.

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
