# Peer review of "Bioengineering for the Microbial Degradation of Petroleum Hydrocarbon Contaminants"

_bioengineering, 2023, doi:10.3390/bioengineering10030347_

Round 1
Reviewer 1 Report
In the paper, the author pointed Bioengineering for Microbial Degradation of Petroleum Hydrocarbon Contaminants. I recommend this article is worthy to publish in the journal after correction of some points described bellow:
1. In my opinion, the term "artificial microbial consortia" is incorrect. A microbial consortium is defined as a group of microorganisms that is species-defined and in which the functions of the bacteria are known. In addition, the bacteria are in physical contact with each other. Therefore, in my opinion, it is better to use the term microbial community. The drawings are very interesting, but not all of their elements are legible. This should be corrected e.g. Fig. 1 2. Titles of figures should be placed under the figure.
3. Chapter 2.5.2.1. Surfactants Producers. Here, the role of biosurfactants was presented unilaterally as a factor supporting the biodegradation process. However, there are known cases where surfactants are not beneficial in the biodegradation process. Add 2-3 citations from this range
4. In the chapter: 2.5.2.2. Nitrogen Providers should include the article:Staninska-Pi˛eta, J.;Czarny, J.; Juzwa,W.;Wolko, Ł.;Cyplik, P.; Piotrowska-Cyplik, A. Dose–Response Effect of Nitrogen on Microbial Community during Hydrocarbon Biodegradation in Simplified Model System. Appl. Sci. 2022, 12, 6012. https://doi.org/ 10.3390/app12126012
Reviewer 2 Report
The manuscript titled ‘‘Bioengineering for Microbial Degradation of Petroleum Hydrocarbon Contaminants ‘‘, is a well-written review paper the topic fits with the journal’s scope. However, the order of subsections is confusing. There are just three sections (Introduction \ Microbial degradation of n-alkane \ Conclusion). And too many subsections that are embedded into the second section. To understand each part, it would be better to increase the number of ''sections'' and decrease of ''subsections''.
Reviewer 3 Report
The manuscript entitled “Bioengineering for Microbial Degradation of Petroleum Hydrocarbon Contaminants” has potentials and the subject is interesting. However, some points should be addressed before final decision.
1. Although the subject has been well defined, the novelties of this review paper is not clear. Please clearly highlight the novelties of the study in the last paragraph of the Introduction.
2. The introduction is not efficient. The subject has been well introduced. However, it is necessary to summarize some of the review papers established in this field and highlight the novelties and contributions of this study over them.
3. The citations in the Introduction is not desirable. Although this is a review paper and it is expected to have lots of references, the citations should be appropriate. For instance, there are seven references in line 57 only by naming genes. Similarly, for lines 60 and 61 where there are nine references! It is better to name all of them a cite a review paper including all of them. This review paper simply ignores all of review papers published in this field especially in recent years.
4. Conclusions section needs a major revision. For review papers, a future direction and/or future perspectives alongside with a deep discussion on the findings are also essential.
5. Although this is a review paper, the lumped references should be avoided, for instance, [114-117] in lines 321, [88-90] in line 303, and etc.
6. It is recommended to mention the research limitations and the gaps of the field should be covered by future studies.
